# Structure of the IL-27 quaternary receptor signaling complex

Nathanael A Caveney[1†], Caleb R Glassman[1,2†], Kevin M Jude[1,3], Naotaka Tsutsumi[1,3], K Christopher Garcia[1,2,3]*

[1]Department of Molecular and Cellular Physiology, Stanford University School of Medicine, Stanford, United States; [2]Program in Immunology, Stanford University School of Medicine, Stanford, United States; [3]Howard Hughes Medical Institute, Stanford University School of Medicine, Stanford, United States

**Abstract** Interleukin 27 (IL-27) is a heterodimeric cytokine that functions to constrain T cell-mediated inflammation and plays an important role in immune homeostasis. Binding of IL-27 to cell surface receptors, IL-27Rα and gp130, results in activation of receptor-associated Janus Kinases and nuclear translocation of Signal Transducer and Activator of Transcription 1 (STAT1) and STAT3 transcription factors. Despite the emerging therapeutic importance of this cytokine axis in cancer and autoimmunity, a molecular blueprint of the IL-27 receptor signaling complex, and its relation to other gp130/IL-12 family cytokines, is currently unclear. We used cryogenic-electron microscopy to determine the quaternary structure of IL-27, composed of p28 and Epstein-Barr Virus-Induced 3 (Ebi3) subunits, bound to receptors, IL-27Rα and gp130. The resulting 3.47 Å resolution structure revealed a three-site assembly mechanism nucleated by the central p28 subunit of the cytokine. The overall topology and molecular details of this binding are reminiscent of IL-6 but distinct from related heterodimeric cytokines IL-12 and IL-23. These results indicate distinct receptor assembly mechanisms used by heterodimeric cytokines with important consequences for targeted agonism and antagonism of IL-27 signaling.

## Editor's evaluation

Cytokines are small protein signaling molecules with a diverse range of activities in inflammation and immune system function. This manuscript reports the cryo-EM structure of the cytokine interleukin-27 (IL-27) bound to soluble domains of two receptor subunits, IL-27Rα and gp130. IL-27 is a composite cytokine consisting of the protein p28 bound to EBI3, which resembles soluble cytokine receptors such as the receptors for IL-6, IL-11 or CNTF. IL-27 signals predominantly via STAT1 and plays an important role in immune homeostasis. The data provide a detailed molecular view of how IL-27 binds to its receptor.

*For correspondence:
kcgarcia@stanford.edu

†These authors contributed equally to this work

## Introduction

Cytokines are secreted factors that mediate cell-cell communication in the immune system (**Spangler et al., 2015**). Binding of cytokines to cell surface receptors, in most cases, leads to activation of receptor-associated Janus Kinase (JAK) proteins, which phosphorylate each other as well as downstream Signal Transducer and Activator of Transcription (STAT) proteins, triggering nuclear translocation and regulation of gene expression. Cytokines can be classified by their use of shared receptors which transduce signals for multiple cytokines within a family. Interleukin-6 signal transducer (IL-6ST), also known as glycoprotein 130 (gp130), is a shared receptor that mediates signaling of multiple cytokines including IL-6, IL-11, and IL-27. Unlike other gp130 family cytokines, IL-27 is a heterodimeric

cytokine consisting of a four-helix bundle, IL-27p28 (p28), with similarity to IL-6, complexed with a secreted binding protein, Epstein-Barr Virus-Induced 3 (Ebi3), with homology to type I cytokine receptors (*Pflanz et al., 2002*). IL-27 signals through a receptor complex consisting of IL-27Rα (TCCR/WSX-1) and gp130 expressed on T cells, NK cells, monocytes, dendritic cells, B cells as well as subsets of endothelial and epithelial cells (*Pflanz et al., 2002*; *Pflanz et al., 2004*; *Heng et al., 2008*; *Figure 1A*). Binding of IL-27 to its receptor subunits triggers the activation of receptor-associated JAK1 and JAK2 leading to phosphorylation of STAT1 and STAT3 (*Lucas et al., 2003*; *Owaki et al., 2008*; *Wilmes et al., 2021*). Functionally, IL-27 signaling serves to constrain inflammation by antagonizing differentiation of pro-inflammatory Th17 cells (*Stumhofer et al., 2006*), stimulating T-bet expression in regulatory T cells (Tregs) (*Hall et al., 2012*), and inducing production of the anti-inflammatory cytokine IL-10 (*Stumhofer et al., 2007*). The important role of IL-27 in constraining inflammation has prompted the development of antagonist antibodies, one of which is currently being developed as a cancer immunotherapy (*Patnaik et al., 2021*).

Although IL-27 is a member of the gp130 family, its heterodimeric composition is similar to IL-12 and IL-23, which are cytokines that share a p40 subunit analogous to Ebi3. In this system, the p40 subunit of IL-12 and IL-23 directly engages the shared receptor IL-12Rβ1 in a manner distinct from IL-6 and related cytokines (*Glassman et al., 2021a*). Given the diverse receptor assembly mechanisms used by cytokine receptors, we sought to determine the structure of IL-27 in complex with its signaling receptors, gp130 and IL-27Rα, by cryoEM.

Here we report the 3.47 Å resolution structure of the complete IL-27 receptor complex. In this structure, the central IL-27 cytokine engages IL-27Rα through a composite interface consisting of p28 and the D2 domain of Ebi3. A conserved tryptophan residue at the tip of p28 then engages the D1 domain of gp130 to facilitate quaternary complex assembly. This receptor assembly mechanism bears striking resemblance to that of IL-6 but is distinct from IL-12 and IL-23, indicating divergent receptor binding modes for heterodimeric cytokines.

## Results
### CryoEM structure of the IL-27 quaternary complex

Interleukin 27 (p28/Ebi3) signals through a receptor complex composed of IL-27Rα and the shared receptor gp130 (*Figure 1A*). The extracellular domain of IL-27Rα consists of five fibronectin type III (FNIII) domains (D1–5) of which the N-terminal D1–D2 constitute a cytokine-binding homology region (CHR). gp130 has a similar domain architecture but with the addition of an N-terminal immunoglobin (Ig) domain (D1). Initial attempts at reconstituting a soluble receptor complex through mixing of IL-27, IL-27Rα D1–D2, and gp130 D1–D3 resulted in dissociation on size-exclusion chromatography due to the low affinity of gp130. To stabilize the complex, we expressed p28 fused to gp130 through a long (20 a.a.) flexible linker which enabled us to purify the complete IL-27 receptor complex by size exclusion chromatography. Importantly, the long linker did not constrain the binding mode, with ~110 Å of unresolved residues spanning a distance of 38 Å between termini (*Figure 1—figure supplement 2*), but rather raised the effective concentration of gp130. The complex was vitrified and subject to single-particle cryoEM analysis. The quaternary complex was determined to a resolution of 3.47 Å (*Figure 1—figure supplement 1* and *Table 1*). All domains of IL-27 and IL-27Rα, as well as D1 of gp130 had well resolved cryoEM density (*Figure 1B-C*, *Figure 1—figure supplement 1E*), with local resolution estimates near and exceeding 3 Å in this region (*Figure 1—figure supplement 1B*). The local resolution for the D2 density of gp130 is lower, yet able to be built with confidence, while D3 density was interpretable for domain placement (*Figure 1C*, *Figure 1—figure supplement 1B*).

The IL-27 quaternary complex exhibits a molecular architecture containing 'sites 1–3', as seen in other gp130 family cytokines (*Boulanger et al., 2003*) and as hypothesized previously (*Skiniotis et al., 2008*; *Figure 1D*). In this structure, IL-27 bridges IL-27Rα and gp130 to initiate downstream signaling through JAK1 and JAK2 (*Ferrao et al., 2016*; *Pradhan et al., 2010*). Within IL-27, the four-helical bundle of p28 packs against the hinge between Ebi3 D1 and D2 in a site 1 interaction. Opposite Ebi3, IL-27Rα D1-D2 engages the helical face of p28 in an interaction stabilized by stem contact between the D2 domains of IL-27Rα and Ebi3 to form a site 2 interaction. On the posterior face of IL-27, the D1 domain of gp130 binds to a conserved tryptophan at the tip of p28 to make a classical site 3 interaction (*Boulanger et al., 2003*).

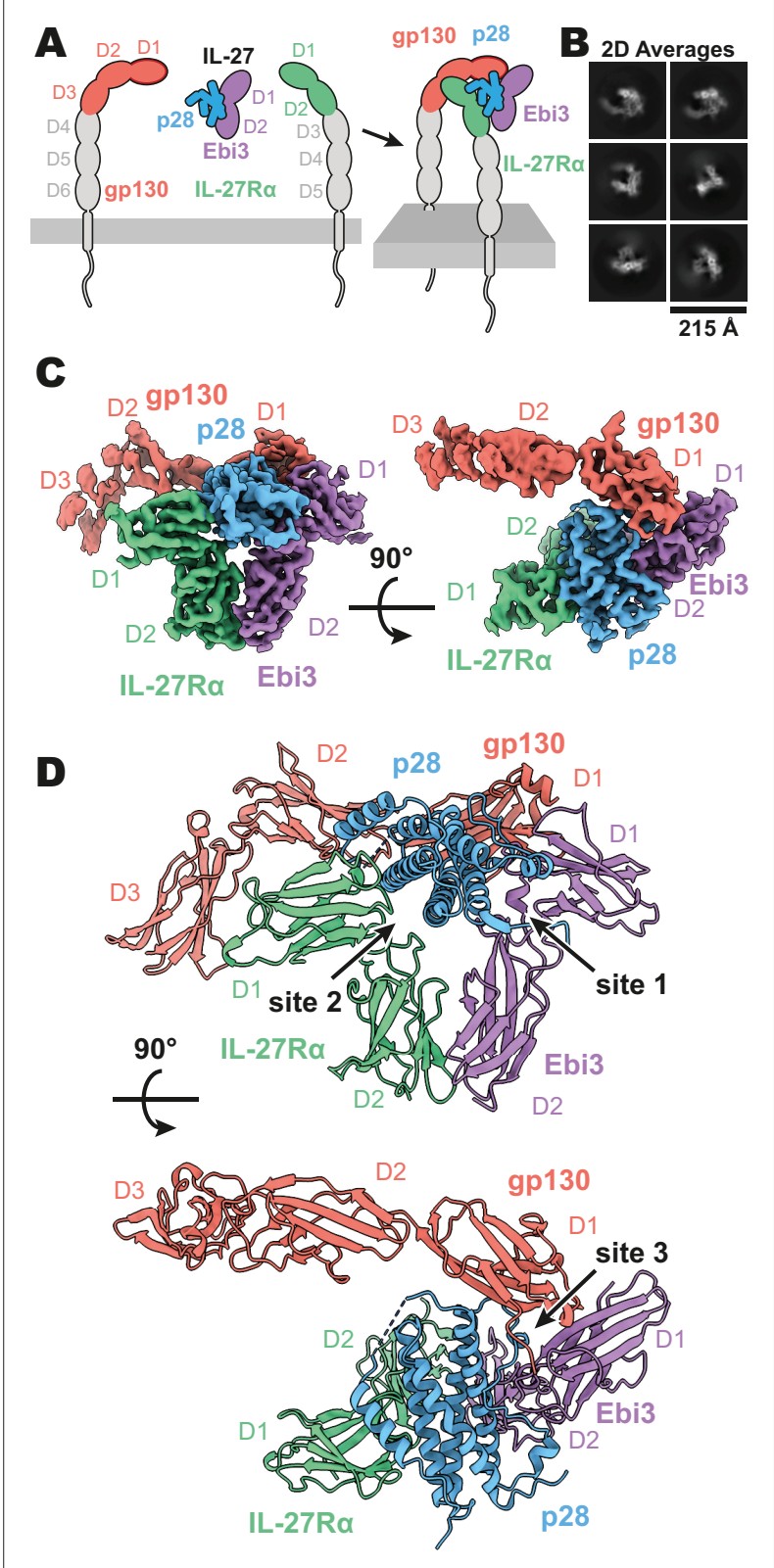

**Figure 1.** Composition and cryogenic-electron microscopy (cryoEM) structure of human interleukin 27 (IL-27) quaternary complex. (**A**) Cartoon representation of the components of the IL-27 quaternary signaling complex. gp130 (red), p28 (blue), Epstein-Barr Virus-Induced 3 (Ebi3; purple), IL-27Rα (green), with domains excluded from the imaged constructs in gray. The D1 Ig domain of gp130 is represented by red, with an additional dark red

*Figure 1 continued on next page*

*Figure 1 continued*

outline to distinguish it from FNIII domains in gp130, Ebi3, and IL-27Rα. (**B**) Reference-free 2D averages from cryoEM of the IL-27 quaternary complex. (**C**) Refined and sharpened cryoEM density maps of IL-27 quaternary complex, colored as in (**A**). (**D**) Ribbon representation of the atomistic modeling of IL-27 quaternary complex, colored as in (**A**).

The online version of this article includes the following figure supplement(s) for figure 1:

**Figure supplement 1.** Interleukin 27 (IL-27) quaternary complex cryogenic-electron microscopy (cryoEM) data processing.

**Figure supplement 2.** Distances and constraints of the Ebi3-gp130 GS-linker.

## Site 1–3 interface architecture of the IL-27 quaternary complex

A site 1 interface between p28 and Ebi3 is used to form the heterodimeric cytokine IL-27. In this interaction, the alpha helices A and D, the AB loop, and the C-terminus of p28 pack tightly against the hinge region between D1 and D2 of Ebi3 (*Figure 2A, B*). This interaction is mediated by a variety of hydrogen bonds between p28 and the hinge region of Ebi3, particularly along alpha helices A and D of p28. At the center of this interface is p28 R219 which is coordinated by D207 and T209 in Ebi3. Extending from this region, a hydrophobic patch of residues on the C-terminus and AB loop of p28 (F94, W97, L223, W232) packs against residues in the Ebi3 hinge (F97, F157, I160). The extensive and hydrophobic nature of this interface is consistent with the observation that co-expression of p28 and Ebi3 is required for efficient cytokine secretion (*Pflanz et al., 2002*).

Opposing the Ebi3-binding site, p28 engages the hinge region of IL-27Rα (*Figure 2C*, site 2 a). This interaction is stabilized by stem-stem contacts between the D2 domains of IL-27Rα and Ebi3 (*Figure 2C*, site 2b). Alpha helices A and C of p28 form the bulk of the site 2a interaction. In contrast to the site 1 interaction which forms the holo-cytokine, the site 2a interaction is more limited both in terms of hydrophobicity and buried surface area (Site 1: 1,339.4 Å$^2$, Site 2a: 810.4 Å$^2$) (*Krissinel and Henrick, 2007*). One notable feature of this interface is the contribution of a 13 amino acid polyglutamic acid region in p28. This region is not well resolved in the cryoEM density, but may form an alpha helix (polyE helix) and contact R74 and K77 in the IL-27Rα D1 domain. Cytokine binding is stabilized by 'stem-stem' interactions between D2 FNIII domains of IL-27Rα and Ebi3. This site 2b interaction is characterized by a high degree of shape and charge complementarity with an arginine-rich patch of Ebi3 (R143, R171, R194) packing against the positively charged base of IL-27Rα D2 domain (D138, D142, E146). The composite nature of IL-27Rα binding explains, in part, the inability of p28 to mediate IL-27 signaling in the absence of Ebi3 (*Stumhofer et al., 2010*).

On the posterior face of the cytokine, a classical site 3 interface is formed through the interaction of the D1 Ig domain of gp130 associating with both p28 (site 3a) and Ebi3 (site 3b) (*Figure 2D, E*). Similar to IL-12 family members, IL-23 and IL-12 (*Glassman et al., 2021a*), and the gp130 family members IL-6 (*Boulanger et al., 2003*) and viral IL-6 (*Chow et al., 2001*), this interaction is anchored by the conserved W197 on p28 which packs tightly against the base of the D1 Ig domain of gp130 (*Figure 2F*). The site 3 interaction is extended by the AB loop of p28 which contacts both gp130 and Ebi3. We do not observe an interaction between the N-terminus of gp130 and helix D of the four-helix bundle as observed for other site 3 interactions; however, this may be due to the use of a flexible GS linker which connects p28 to gp130. The tip of gp130 D1 packs against the top of Ebi3 D1 in a limited interface centered around Ebi3 F118.

The heterodimeric nature of IL-27 has led some to classify it as an IL-12 family cytokine (*Trinchieri et al., 2003*). However, comparison of the IL-27 receptor complex with that of IL-23 and IL-12 reveals striking differences (*Figure 3A–C*). In IL-27, the central p28 subunit engages all receptor components. In contrast, each subunit of IL-23 and IL-12 engages a different receptor in a modular interaction mechanism (*Glassman et al., 2021a*). The assembly of IL-27 more closely resembles that of IL-6, in which the central four-helix bundle encodes binding sites for all receptor components (*Figure 3D*). However, in the case of IL-6, this motif is duplicated through a C2 symmetry axis due to a dual role of gp130 at site 2 and site 3. The similarity between IL-27 and IL-6 is observed not only in overall architecture but also in molecular detail, where IL-6, vIL-6, and IL-27 all engage gp130 using a highly convergent interface in which a tryptophan from helix D of the cytokine (W197 in p28, W185 in IL-6, W166 in vIL-6) forms an aromatic anchor that is capped by Y116 of gp130 (*Figures 2F and 3D,*

**Table 1.** CryoEM data collection, refinement, and validation statistics.

| | IL-27 Complex (PDB 7U7N/EMD-26382) |
|---|---|
| **Data collection and processing** | |
| Magnification | 105,000 |
| Voltage (keV) | 300 |
| Electron exposure (e⁻/Å²) | 60 |
| Defocus range (μm) | –0.8 to –2.0 |
| Pixel size (Å) | 0.839 |
| Symmetry imposed | C1 |
| Initial particle images | 6,387,370 |
| Final particle images | 548,147 |
| Map resolution FSC threshold (Å) | 0.143 |
| Map resolution (Å) | 3.47 |
| | |
| **Refinement** | |
| Initial model used (PDB) | AlphaFold |
| Model resolution FSC threshold (Å) | 0.143 |
| Model resolution (Å) | 1.9 |
| Map sharpening $B$-factor (Å²) | 189.6 |
| Model Composition | |
| Non-hydrogen atoms | 7,284 |
| Protein residues | 884 |
| Ligands | 16 |
| $B$-factors (Å²) | |
| Protein | 102.97 |
| Ligand | 114.62 |
| R.m.s. deviations | |
| Bond lengths (Å) | 0.003 |
| Bond angles (°) | 0.610 |
| Validation | |
| MolProbity score | 1.60 |
| Clashscore | 8.81 |
| EMringer score | 2.33 |
| Rotamer outliers (%) | 0.89 |
| Ramachandran plot | |
| Favored (%) | 97.37 |
| Allowed (%) | 2.63 |
| Outliers (%) | 0.00 |

E). Alignment of gp130 D1–(v)IL-6 pairs to the IL-27 structure reveals a tight correlation in gp130 binding pose with Cα RMSDs of 1.039 Å (IL-6) and 1.015 Å (vIL-6) across 149 and 124 trimmed residue pairs. This striking convergence of gp130 binding modes helps to explain the ability of p28 to antagonize IL-6 signaling (*Stumhofer et al., 2010*) and contextualizes its place in respect to both gp130 and IL-12 family cytokines.

## Discussion

A key paradigm in cytokine structural biology is the use of shared receptor components that mediate diverse signaling outputs. IL-27 extends this, given that it shares the gp130 subunit with IL-6 but mediates distinct and often counter-vailing functional effects. Here we find that IL-27 assembles a receptor complex reminiscent of IL-6 but distinct from other heterodimeric IL-12/23 class cytokines. Interestingly, the soluble form of IL-6Rα (sIL-6Rα) generated through alternative splicing or proteolytic cleavage can complex with IL-6 to potentiate signaling in gp130-expressing cells (*Kishimoto and Kang, 2022*). This IL-6/sIL-6Rα complex thus may be functionally analogous to heterodimeric cytokine IL-27, albeit with gp130 playing the role of both site 2 and 3 receptors in the case of IL-6/sIL-6Rα signalling, allowing for broader range of cells which can be stimulated.

In addition to their roles in IL-27 signaling, p28 and Ebi3 have been implicated in the assembly of alternate cytokine complexes. Ebi3 has been reported to complex with the p35 subunit of IL-12 to form IL-35 (*Collison et al., 2007*) and the p19 subunit of IL-23 to form IL-39 (*Wang et al., 2016*), while p28 has been proposed to interact with IL-6Rα (*Garbers et al., 2013*) and the p40 subunit of IL-12/23 (*Wang et al., 2012*). These results suggest structural plasticity among heterodimeric cytokines; however, additional work is needed to characterize the physiological relevance and biochemical basis for these cytokines.

As in other types of cytokine signaling, there is context-dependent therapeutic potential in both the inhibition and potentiation of IL-27 signaling. The targeting of the heterodimeric cytokines of the IL-12 family has been well explored, with various clinically approved inhibitors, targeting either IL-23 signaling via p19 (Risankizumab, Guselkumab, Tildrakizumab) (*Tait Wojno et al., 2019*) or IL-12 and IL-23 signaling via the shared p40 (Ustekinumab) (*Luo et al., 2010*). In these cases, the inhibition of IL-12 family members antagonizes the pro-inflammatory effects of these cytokines. Due to the role of IL-27 in the regulation T cell-mediated inflammation (*Tait Wojno et al., 2019*), inhibition of IL-27 is currently being explored for its use in the inhibition of aberrant IL-27 signaling in

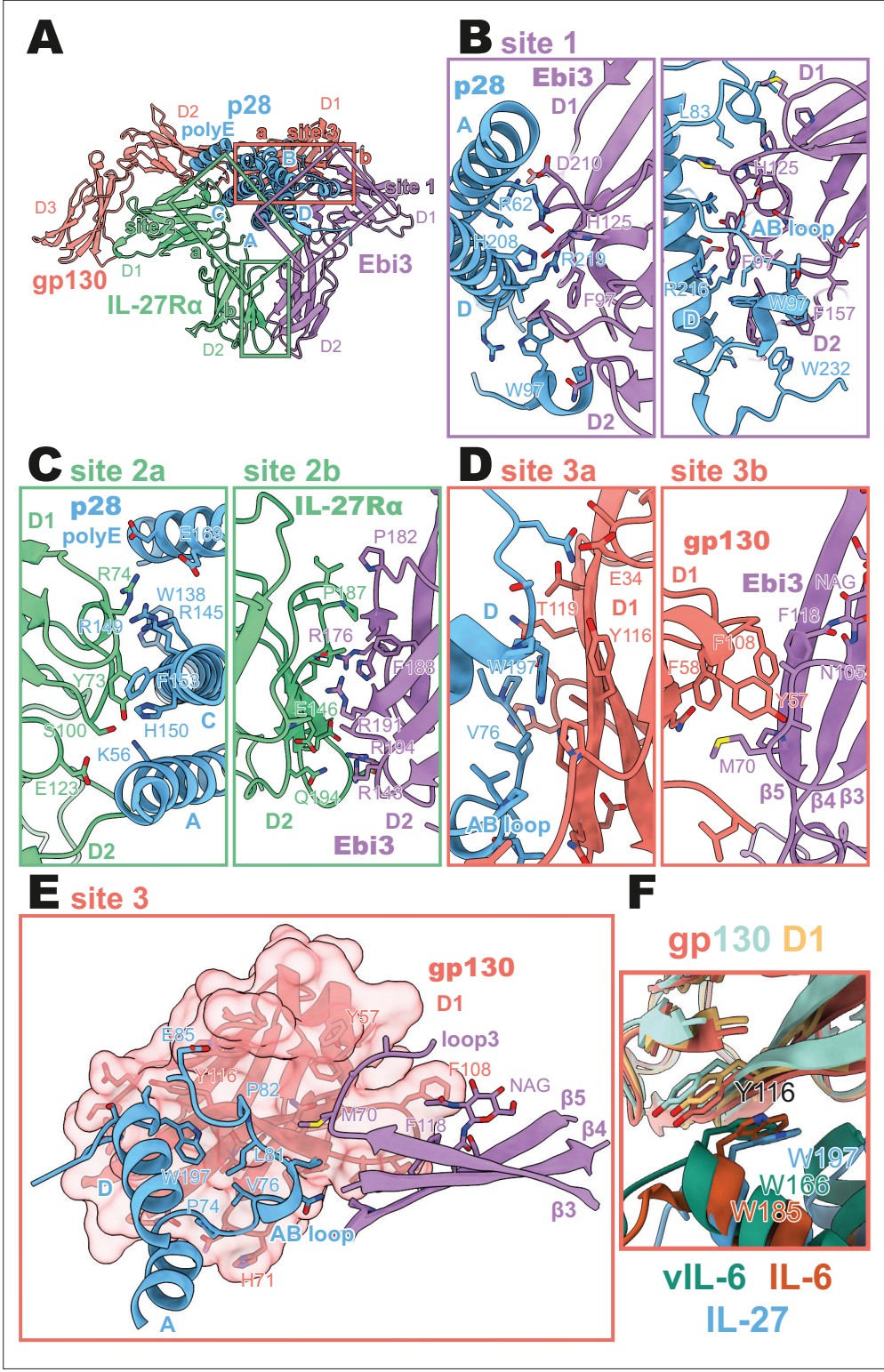

**Figure 2.** Binding interfaces of the human interleukin 27 (IL-27) quaternary complex. (**A**) Ribbon representation of the IL-27 quaternary signaling complex, containing gp130 (red), p28 (blue), Epstein-Barr Virus-Induced 3 (Ebi3; purple), and IL-27Rα (green). Regions containing sites 1, 2, and 3 are boxed in purple, green, and red, respectively. (**B**) Two views of the site 1 interface, colored as in (**A**). (**C**) The site 2 interface, composed of a site 2a, p28 to IL-27Rα, interaction, and a site 2b, Ebi3 to IL-27Rα, interaction. All proteins are colored as in (**A**). (**D,E**) The site 3 interface, composed of a site 3a, p28 to gp130, interaction, and a site 3b, Ebi3 to IL-27Rα, interaction. All proteins

*Figure 2 continued on next page*

*Figure 2 continued*

are colored as in (**A**). (**F**) Structural overlay of site 3 interacting domains from IL-27 complex, IL-6 complex (PDB 1P9M), and viral IL-6 (vIL-6) complex (PDB 1L1R). IL-27 quaternary complex colored as in (**A**), IL-6 ternary complex in orange (IL-6) and yellow (gp130), and vIL-6 binary complex in green (vIL-6) and teal (gp130).

The online version of this article includes the following figure supplement(s) for figure 2:

**Figure supplement 1.** Cryogenic-electron microscopy (CryoEM) density of binding interfaces of the human IL-27 quaternary complex.

cancer (*Patnaik et al., 2021*). The structure of the IL-27 quaternary signaling complex provides ample opportunities for the design of IL-27 inhibitors targeting different steps in receptor assembly to better regulate IL-27 signaling.

On the other end of the spectrum, IL-27 agonism has been explored for its therapeutic use in inflammatory autoimmune dysregulation such as experimental autoimmune encephalomyelitis (*Fitzgerald et al., 2013*) and colitis (*Hanson et al., 2014*). Despite the promise of IL-27 as a therapeutic, the bipartite nature of IL-27 limits its usefulness in the clinic. Using the interfacial information provided in the structure of the IL-27 quaternary complex, structure-guided protein engineering techniques can now be used to improve the therapeutic potential of IL-27. One such application may be to generate single agent IL-27 agonists by affinity maturing the interactions between p28 and its receptor subunit, as has been done recently to generate IL-6 which do not require IL-6Rα binding for efficient signaling (*Martinez-Fabregas et al., 2019*). Conversely, attenuating the affinity of p28 for IL-27Rα might be used to generate cell-type biased partial agonists with selective activity based on differences in IL-27Rα expression across cell-type and activation state (*Villarino et al., 2005*) as has been done for IL-2 (*Glassman et al., 2021b*), IL-10 (*Saxton et al., 2021*), and IL-12 (*Glassman et al., 2021a*).

Resolution of the complete IL-27 receptor complex reveals how this therapeutically important cytokine engages its receptor subunits. The receptor assembly mechanism bears striking resemblance to that of IL-6 but is distinct from IL-12 and IL-23, indicating divergent receptor binding modes for heterodimeric cytokines. This structural insight further paves the way for the continued development of therapeutics that modulate IL-27 signaling.

# Materials and methods

## Key resources table

| Reagent type (species) or resource | Designation | Source or reference | Identifiers | Additional information |
|---|---|---|---|---|
| Cell line (*Homo sapiens*) | Human embryonic kidney cells | GIBCO | Expi293 | |
| Recombinant DNA reagent | pD649-IL-27Rα (plasmid) | This paper | | See: Methods - Cloning and protein expression |
| Recombinant DNA reagent | pD649-Ebi3 (plasmid) | This paper | | See: Methods - Cloning and protein expression |
| Recombinant DNA reagent | pD649-p28-gp130 (plasmid) | This paper | | See: Methods - Cloning and protein expression |
| Software, algorithm | Data collection software | SerialEM | SerialEM | |
| Software, algorithm | Data processing software | Structura Biotechnology Inc. | cryoSPARC | |
| Software, algorithm | Data sharpening software | *Sanchez-Garcia et al., 2021* | DeepEMhancer | |
| Software, algorithm | Initial modeling software | *Jumper et al., 2021* | AlphaFold | |
| Software, algorithm | Graphics software | *Pettersen et al., 2021* | UCSF ChimeraX | |
| Software, algorithm | Modeling and refinement software | *Adams et al., 2010* | Phenix | |
| Software, algorithm | Modeling and refinement software | *Emsley and Cowtan, 2004* | Coot | |
| Software, algorithm | Model validation software | *Barad et al., 2015* | EMRinger | |
| Software, algorithm | Model validation software | *Chen et al., 2010* | MolProbity | |

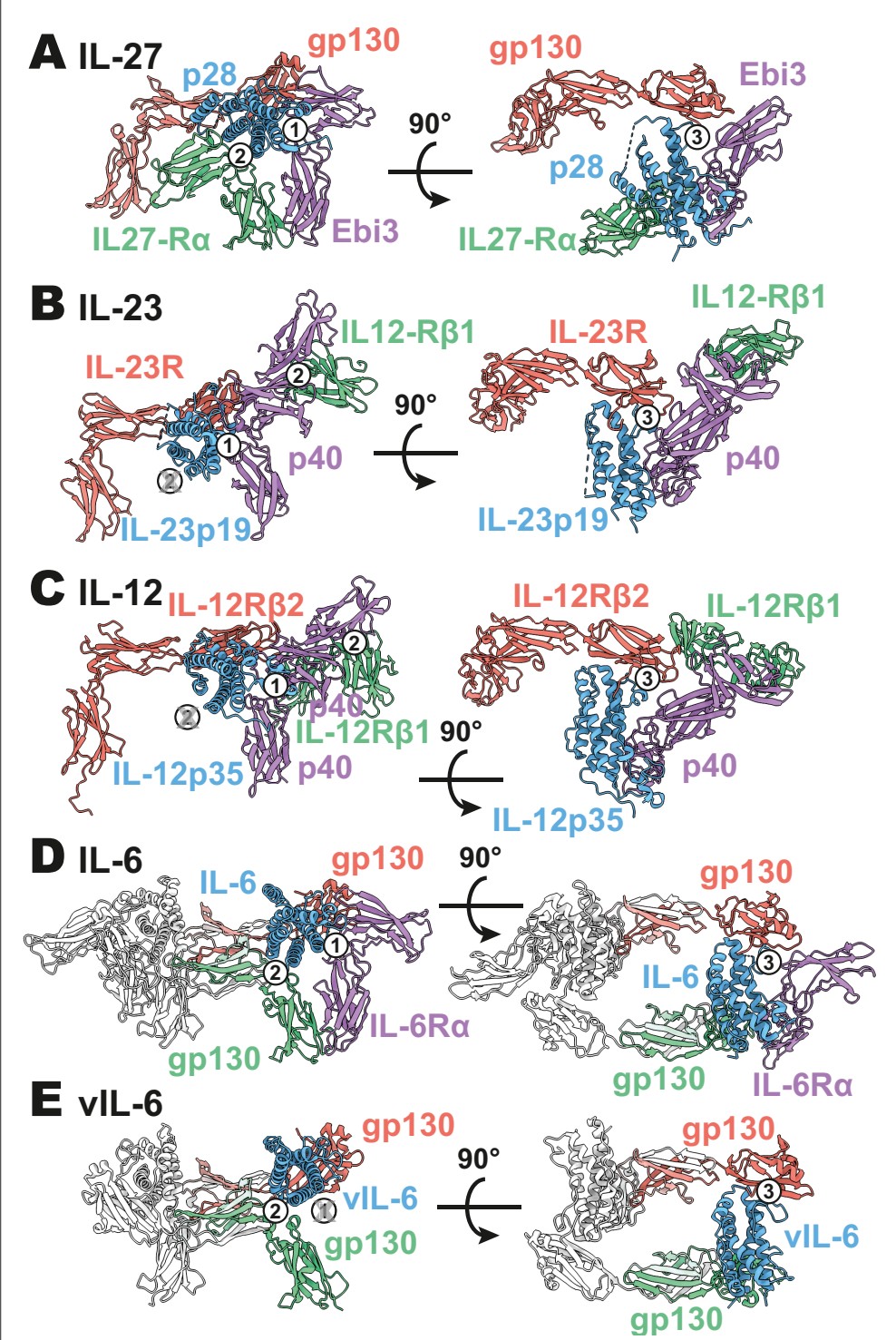

**Figure 3.** Comparison of IL-27 to IL-12 and IL-6 family complexes. (**A**) Ribbon representation of the IL-27 signaling complex, containing gp130 (red), p28 (blue), Epstein-Barr Virus-Induced 3 (Ebi3; purple), and IL-27Rα (green). Sites 1, 2, and 3 are noted with a numbered circle. (**B**) Ribbon representation of the IL-23 signaling complex (IL-12 family), containing IL-23R (red), IL-23p19 (blue), p40 (purple), and IL-12Rβ1 (green) (PDB 6WDQ). Sites 1 and 3 are noted as in (**A**), with the unoccupied site 2 marked with a gray 'X' and the distinct IL-12/IL-23 site 2 marked with a circled number 2. (**C**) Ribbon representation of a model of the IL-12 signaling complex (IL-12 family), containing IL-12Rβ2 (red), IL-12p35 (blue), p40 (purple), and IL-12Rβ1 (green) (**Baek et al., 2021**; **Glassman et al., 2021a**). Sites 1 and 3 are noted as in (**A**), with the unoccupied site 2 marked with a gray 'X' and the distinct IL-12/IL-23

*Figure 3 continued on next page*

*Figure 3 continued*

site 2 marked with a circled number 2. (**D**) Ribbon representation of the IL-6 signaling complex (IL-6 family), containing gp130 (red and green), IL-6 (blue), and IL-6Rα (purple) (PDB 1P9M). Secondary copies of each protein in the complex are colored in white. Sites 1, 2, and 3 are noted as in (**A**). (**E**) Ribbon representation of the viral IL-6 signaling complex (IL-6 family), containing gp130 (red and green) and (blue) (PDB 1I1R). Secondary copies of each protein in the complex are colored in white. Sites 2 and 3 are noted as in (**A**), with the unoccupied site 1 marked with a gray 'X'.

## Cloning and protein expression

Human IL-27Rα (D1-D2, residues 36–231), Ebi3 (residues 21–228), and p28 (residues 29–243)-(GGGGS)$_4$-gp130 (D1-D3, residues 23–321) were cloned into the expression vector pD649. All components were expressed with signal sequence of influenza hemagglutinin and IL-27Rα and Ebi3 contained C-terminal 6-His tags for affinity purification. Proteins were co-expressed in Expi293F cells (GIBCO) maintained in Expi293 Expression Media (GIBCO) at 37 °C with 5% $CO_2$ and gentle agitation.

### Material availability

The expression plasmids for the IL-27 complex are available from KCG (https://kcgarcia@stanford.edu) by request.

## Protein purification

IL-27 quaternary complex was purified by Ni-NTA chromatography followed by size exclusion chromatography with a Superdex 200 column (GE Lifesciences). Fractions containing pure IL-27 complex were pooled and stored at 4°C until vitrification.

## Cryo-electron microscopy

Aliquots of 3 µL of IL-27 quaternary complex were applied to glow-discharged Quantifoil (1.2/1.3) grids. The grids were blotted for 3 s at 100% humidity with an offset of 3 and plunge frozen into liquid ethane using a Vitrobot Mark IV (Thermo Fisher). Grid screening and preliminary dataset collection occurred at Stanford cEMc. Final grids were imaged on a 300 kV FEI Titan Krios microscope (Thermo Fisher) located at the HHMI Janelia Research Campus and equipped with a K3 camera and energy filter (Gatan). Movies were collected at a magnification of ×105,000, corresponding to a 0.839 Å per physical pixel. The dose was set to a total of 50 electrons per Å$^2$ over an exposure of 50 frames. Automated data collection was carried out using SerialEM with a nominal defocus range set from –0.8 to –2.0 µM. 18,168 movies were collected.

## Image processing

All processing was performed in cryoSPARC (*Punjani et al., 2017*) unless otherwise noted (*Figure 1— figure supplement 1*). The 18,168 movies were motion corrected using patch motion correction and micrographs were binned to 0.839 Å per pixel. The contrast transfer functions (CTFs) of the flattened micrographs were determined using patch CTF and 6,387,370 particles were picked using blob picking and subsequently template picking. A subset of 2,008,987 particles were used in reference-free 2D classification. A particle stack containing 109,536 2D cleaned particles was used to generate three ab initio models, one of which resembled a complete complex. This ab initio model was then used against two junk classes in six rounds of iterative heterogenous refinement to reduce the full particle stack to 548,147 particles. These particles were refined using cryoSPARC non-uniform refinement (*Punjani et al., 2020*) followed by local refinement with a mask excluding gp130 D3 domain to achieve a resolution of 3.47 Å. Resolution was determined at a criterion of 0.143 Fourier shell correlation gold-standard refinement procedure. The final map was sharpened using deepEMhancer (*Sanchez-Garcia et al., 2021*).

## Model building and refinement

AlphaFold models (*Jumper et al., 2021*) of IL-27Rα, Ebi3, p28, and gp130 were docked into the map using UCSF Chimera X (*Pettersen et al., 2021*). The resultant model was then refined using Phenix

real space refine (*Adams et al., 2010*) and manual building in Coot (*Emsley and Cowtan, 2004*). The final model fit the map well (EMRinger [*Barad et al., 2015*] score 2.33) and produced a favorable MolProbity score of 1.60 (*Chen et al., 2010*) with side-chain rotamers occupying 97.37% Ramachandran favored and 0.00% outliers (*Table 1*).

## Acknowledgements

We thank Rui Yan at the HHMI Janelia CryoEM Facility for help in microscope operation and final data collection. We thank Liz Montabana and Stanford cEMc for microscope access for preliminary data collection. NAC is a CIHR postdoctoral fellow. KCG is an investigator with the Howard Hughes Medical Institute. KCG is supported by National Institutes of Health grant R01-AI51321, the Mathers Foundation, and the Ludwig Foundation.

## Additional information

### Competing interests

K Christopher Garcia: is the founder of Synthekine. The other authors declare that no competing interests exist.

### Funding

| Funder | Grant reference number | Author |
|---|---|---|
| Canadian Institutes of Health Research | | Nathanael A Caveney |
| Howard Hughes Medical Institute | | K Christopher Garcia |
| National Institute of Allergy and Infectious Diseases | R01-AI51321 | K Christopher Garcia |

The funders had no role in study design, data collection and interpretation, or the decision to submit the work for publication.

### Author contributions

Nathanael A Caveney, Caleb R Glassman, Conceptualization, Data curation, Formal analysis, Investigation, Methodology, Validation, Visualization, Writing - original draft, Writing – review and editing; Kevin M Jude, Formal analysis, Writing – review and editing; Naotaka Tsutsumi, Investigation, Methodology, Writing – review and editing; K Christopher Garcia, Conceptualization, Funding acquisition, Project administration, Supervision, Writing – review and editing

### Author ORCIDs

Nathanael A Caveney ![ORCID]http://orcid.org/0000-0003-4828-3479
Caleb R Glassman ![ORCID]http://orcid.org/0000-0002-3342-7989
Kevin M Jude ![ORCID]http://orcid.org/0000-0002-3675-5136
Naotaka Tsutsumi ![ORCID]http://orcid.org/0000-0002-3617-7145
K Christopher Garcia ![ORCID]http://orcid.org/0000-0001-9273-0278

### Decision letter and Author response

Decision letter https://doi.org/10.7554/eLife.78463.sa1
Author response https://doi.org/10.7554/eLife.78463.sa2

## Additional files

### Supplementary files

• MDAR checklist

## Data availability

CryoEM maps and atomic coordinates for human IL-27 quaternary complex have been deposited in the EMDB (EMD- 26382) and PDB (7U7N) respectively.

The following datasets were generated:

| Author(s) | Year | Dataset title | Dataset URL | Database and Identifier |
|---|---|---|---|---|
| Caveney NA, Glassman CR, Jude KM, Tsutsumi N, Garcia KC | 2022 | IL-27 quaternary receptor signaling complex | https://www.rcsb.org/structure/7U7N | RCSB Protein Data Bank, 7U7N |
| Caveney NA, Glassman CR, Jude KM, Tsutsumi N, Garcia KC | 2022 | IL-27 quaternary receptor signaling complex | https://www.ebi.ac.uk/emdb/EMD-26382 | Electron Microscopy Data Bank, EMD-26382 |

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
