## [Editor Report]

Cytokines are small protein signaling molecules with a diverse range of activities in inflammation and immune system function. This manuscript reports the cryo-EM structure of the cytokine interleukin-27 (IL-27) bound to soluble domains of two receptor subunits, IL-27Rα and gp130. IL-27 is a composite cytokine consisting of the protein p28 bound to EBI3, which resembles soluble cytokine receptors such as the receptors for IL-6, IL-11 or CNTF. IL-27 signals predominantly via STAT1 and plays an important role in immune homeostasis. The data provide a detailed molecular view of how IL-27 binds to its receptor.

---

## [Decision Letter]

**Decision letter after peer review:**

Thank you for submitting your article "Structure of the IL-27 quaternary receptor signaling complex" for consideration by *eLife*. Your article has been reviewed by 3 peer reviewers, and the evaluation has been overseen by a Reviewing Editor and Volker Dötsch as the Senior Editor. The reviewers have opted to remain anonymous.

Essential revisions:

Please address the points raised by reviewers regarding clarity and presentation as well as referencing and discussion of some prior work. No additional data is required in a revised manuscript.

*Reviewer #1 (Recommendations for the authors):*

In the opening paragraph of the introduction the authors indicate that when cytokines bind their receptors this leads to the activation of JAK-STAT signaling. While that is the case for the cytokines discussed here, it is not a general characteristic of all cytokines, which is how this statement currently reads.

Also, given the way that the two receptor subunits bend to facilitate binding of IL-27, it would be useful to see how these domains connect to the rest of the receptor domains on the surface of the plasma membrane to give a better visual of the overall interaction of IL-27 with the membrane bound receptor. The authors provided a figure such as this in their similar paper describing the interaction of IL-6 with its receptor (Boulanger et al., Science 2003). A figure or an additional figure panel such as this would also be useful to the reader here.

*Reviewer #2 (Recommendations for the authors):*

This first structure of the complex of the heterodimeric cytokine IL-27 bound to the extracellular portions of the two receptor subunits gp130 and WSX-1 is very interesting and will be a major breakthrough on the way to fully understand the structural plasticity of cytokines and their cognate receptor complexes. EBI3 is part of the cytokines IL-27, IL-35 and IL-39. P28 has been shown to signal via EBI3 and via the IL-6R, which interestingly results in different receptor complexes and different intracellular signals. To fully explain this structural plasticity, a higher resolution structure will be required, which can be expected in the future.

There are some points the authors might want to address.

1. The author state that the receptor protein is mainly expressed on T-cells and NK cells. This is not fully correct since WSX-1 is also expressed on monocytic cells, dendritic cells, B-cells, endothelial cells and some epithelial cells. This should be corrected.

2. The authors describe that the complex of IL-27 and the two receptor proteins was made possible by covalently fusing p28 to gp130. Some more details and a scheme should be given to illustrate this approach. The authors argue that the linker did not constrain the binding mode. The experiments on which this statement was based should be shown in the supplemental material.

3. Along the same line: the authors state that the site 3 binding between p28 and the D1 domain of gp130 is similar to the site 3 binding of human or viral IL-6. In the case of IL-6, however, it has been demonstrated that the NH2-terminus of gp130 inserts and interacts with the tip of the cytokine. Is such an interaction also possible for p28 and would such an interaction be hindered by the covalent linker? This point should be mentioned and discussed.

4. EBI3 is part of the cytokines IL-27, IL-35 and IL-39. P28 has been shown to signal via EBI3 and via the IL-6R, which interestingly results in different receptor complexes and different intracellular signals. This should at least be mentioned (and cited) in the discussion and the problem of structural plasticity should be explained to the reader.

*Reviewer #3 (Recommendations for the authors):*

Suggestion: use the term 'local resolution' instead of 'localized resolution'.

line 89: 'electron density is not the correct term for maps determined using cryoEM. use 'cryoEM density or 'density'.

Figure 1:

It would be useful if the schematic in panel A had the proteins arranged with their binding interfaces oriented in the way that would be to interact with each other in the tetrameric complex.

Figure S1:

A: What part of the map is locally refined (and why)? Showing the mask(s) used would be helpful.

Figure S2: make the boxes in panel A more representative (i.e. squares) of the zoom-in area in panels B-D.

Map + Model:

When the map is deposited, it is requested that an unsharpened map is deposited in addition to the DeepEMhancer sharped map since CNNs trained like this could potentially invent misleading protein features (and since it is trained on real protein features, unlike previous generation sharpening algos, the 'fake' features will look 'real').

I don't think the density in the provided structure supports a polyE helix in p28. Maybe it should not be modeled?

The map used to estimate local resolution in figure S1B should not be the DeepEMhancer-sharped (maybe it is not, but unclear from the text which map was used, and I'm surprised that the local resolution is estimated to be so similar in different areas).

---

## [Author Response]

Reviewer #1 (Recommendations for the authors):In the opening paragraph of the introduction the authors indicate that when cytokines bind their receptors this leads to the activation of JAK-STAT signaling. While that is the case for the cytokines discussed here, it is not a general characteristic of all cytokines, which is how this statement currently reads.

We have adjusted this statement to read:

“Binding of cytokines to cell surface receptors, in most cases, leads to activation of receptor-associated Janus Kinase (JAK) proteins which phosphorylate each other as well as downstream Signal Transducer and Activator of Transcription (STAT) proteins, triggering nuclear translocation and regulation of gene expression.”

Also, given the way that the two receptor subunits bend to facilitate binding of IL-27, it would be useful to see how these domains connect to the rest of the receptor domains on the surface of the plasma membrane to give a better visual of the overall interaction of IL-27 with the membrane bound receptor. The authors provided a figure such as this in their similar paper describing the interaction of IL-6 with its receptor (Boulanger et al., Science 2003). A figure or an additional figure panel such as this would also be useful to the reader here.

We have added a panel to Figure 1.

Reviewer #2 (Recommendations for the authors):This first structure of the complex of the heterodimeric cytokine IL-27 bound to the extracellular portions of the two receptor subunits gp130 and WSX-1 is very interesting and will be a major breakthrough on the way to fully understand the structural plasticity of cytokines and their cognate receptor complexes. EBI3 is part of the cytokines IL-27, IL-35 and IL-39. P28 has been shown to signal via EBI3 and via the IL-6R, which interestingly results in different receptor complexes and different intracellular signals. To fully explain this structural plasticity, a higher resolution structure will be required, which can be expected in the future.There are some points the authors might want to address.1. The author state that the receptor protein is mainly expressed on T-cells and NK cells. This is not fully correct since WSX-1 is also expressed on monocytic cells, dendritic cells, B-cells, endothelial cells and some epithelial cells. This should be corrected.

We have updated the text which now reads:

“IL-27 signals through a receptor complex consisting of IL-27Rα (TCCR/WSX-1) and gp130 expressed on T cells, NK cells, monocytes, dendritic cells, B cells as well as subsets of endothelial and epithelial cells”

2. The authors describe that the complex of IL-27 and the two receptor proteins was made possible by covalently fusing p28 to gp130. Some more details and a scheme should be given to illustrate this approach. The authors argue that the linker did not constrain the binding mode. The experiments on which this statement was based should be shown in the supplemental material.

We have adjusted the description of this as follows:

“Importantly, the long linker did not constrain the binding mode, with ~110 Å of unresolved residues spanning a distance of 38 Å between termini (Figure 1 – Supplement 3), but rather raised the effective concentration of gp130.” And added an additional figure supplement to Figure 1 (Figure 1 – supplement 3).

3. Along the same line: the authors state that the site 3 binding between p28 and the D1 domain of gp130 is similar to the site 3 binding of human or viral IL-6. In the case of IL-6, however, it has been demonstrated that the NH2-terminus of gp130 inserts and interacts with the tip of the cytokine. Is such an interaction also possible for p28 and would such an interaction be hindered by the covalent linker? This point should be mentioned and discussed.

We have elaborated on the placement of the AB loop to include discussion of the N-terminus of gp130, as follows:

“The site 3 interaction is extended by the AB loop of p28 which contacts both gp130 and Ebi3. We do not observe an interaction between the N-terminus of gp130 and helix D of the four-helix bundle as observed for other site 3 interactions, however, this may be due to the use of a flexible GS linker which connects p28 to gp130.”

4. EBI3 is part of the cytokines IL-27, IL-35 and IL-39. P28 has been shown to signal via EBI3 and via the IL-6R, which interestingly results in different receptor complexes and different intracellular signals. This should at least be mentioned (and cited) in the discussion and the problem of structural plasticity should be explained to the reader.

A paragraph discussing alternative heterodimeric cytokines has been added to the discussion:

“In addition to their roles in IL-27 signaling, p28 and Ebi3 have been implicated in the assembly of alternate cytokine complexes. Ebi3 has been reported to complex with the p35 subunit of IL-12 to form IL-35 (Collison et al., 2007) and the p19 subunit of IL-23 to form IL-39 (Wang et al., 2016) while p28 has been proposed to interact with IL-6Rα (Garbers et al., 2013) and the p40 subunit of IL-12/23 (Wang et al., 2012). These results suggest structural plasticity among heterodimeric cytokines, however, additional work is needed to characterize the physiological relevance and biochemical basis for these cytokines.”

Reviewer #3 (Recommendations for the authors):Suggestion: use the term 'local resolution' instead of 'localized resolution'.line 89: 'electron density is not the correct term for maps determined using cryoEM. use 'cryoEM density or 'density'.

We have changed all instances of “localised resolution” to “local resolution” and all instances of “electron density” to “density”

Figure 1:It would be useful if the schematic in panel A had the proteins arranged with their binding interfaces oriented in the way that would be to interact with each other in the tetrameric complex.

We have adjusted the orientation of the proteins in F1A.

Figure S1:A: What part of the map is locally refined (and why)? Showing the mask(s) used would be helpful.

Alignment including the D3 domain of gp130 resulted in poorer global resolutions, so the complex was locally refined with a mask excluding this domain. The mask is now depicted in Figure 1 – supplement 1. We have clarified this as follows in the methods:

“These particles were refined using cryoSPARC non-uniform refinement (Punjani et al., 2020) followed by local refinement with a mask excluding gp130 D3 domain to achieve a resolution of 3.47 Å.”

Figure S2: make the boxes in panel A more representative (i.e. squares) of the zoom-in area in panels B-D.

We have adjusted the boxes to be more representative of the zoom-in regions.

Map + Model:When the map is deposited, it is requested that an unsharpened map is deposited in addition to the DeepEMhancer sharped map since CNNs trained like this could potentially invent misleading protein features (and since it is trained on real protein features, unlike previous generation sharpening algos, the 'fake' features will look 'real').

We have deposited the map with both unsharpened half-maps.

I don't think the density in the provided structure supports a polyE helix in p28. Maybe it should not be modeled?

We agree that the density is weaker in the region of the polyE helix, likely due to local flexibility. We have adjusted our description of the polyE region and its interaction to account for this uncertainty:

“One notable feature of this interface is the contribution of a 13 amino acid polyglutamic acid region in p28. This region is not well observed in the cryoEM density, but may form an α helix (polyE helix) and contact R74 and K77 in the IL-27Rɑ D1 domain.”

The map used to estimate local resolution in figure S1B should not be the DeepEMhancer-sharped (maybe it is not, but unclear from the text which map was used, and I'm surprised that the local resolution is estimated to be so similar in different areas).

The map presented was the DeepEMhanced map, the figure is now changed to have the globally sharpened map for the local resolution estimation. As well, the gradient range used to color the map has been adjusted to better depict the values present.